# Effects of IGF1 rs6214 Polymorphism and Milk Consumption on Serum Levels of IGF-1 and GH and Body Composition

**DOI:** 10.3390/metabo15080556

**Published:** 2025-08-20

**Authors:** Julio Cesar Grijalva-Avila, Ignacio Villanueva-Fierro, Sandra Consuelo Martínez-Estrada, Gerardo Grijalva-Avila, Alonso Gándara-Mireles, Gildardo Rivera, Antonio Loera-Castañeda, Horacio Almanza-Reyes, Leslie Patrón-Romero, Verónica Loera-Castañeda

**Affiliations:** 1Laboratorio de Farmacogenómica y Biomedicina Molecular, CIIDIR Unidad Durango, Instituto Politécnico Nacional, Durango 34220, Mexico; jcgrijalva69@gmail.com (J.C.G.-A.); ivillanuevaf@ipn.mx (I.V.-F.); sandramartinez@cbtis130.edu.mx (S.C.M.-E.); alonso_930@hotmail.com (A.G.-M.);; 2Red Latinoamericana de Implementación y Validación de Guías Clínicas Farmacogenómicas (RELIVAF-CYTED), 28015 Madrid, Spain; 3Centro de Bachillerato Tecnológico Industrial y de Servicios (CBTIS)130, Durango 34170, Mexico; 4Ingeniería en Tecnologías de Manufactura Avanzada, Universidad Politécnica de Durango, Durango 34307, Mexico; 5Laboratorio de Biotecnología Farmacéutica, Centro de Biotecnología Genómica, Instituto Politécnico Nacional, Reynosa 88710, Mexico; gildardors@hotmail.com; 6PRECIPHARM, Medicina Personalizada y de Precisión, Durango 34080, Mexico; 7Faculty of Medicine and Psychology, Autonomous University of Baja California, Tijuana 22390, Mexico; almanzareyes@uabc.edu.mx (H.A.-R.); leslie.patron@uabc.edu.mx (L.P.-R.)

**Keywords:** body composition, IGF1, milk intake, rs6214 variants, IGF-1 and GH concentrations

## Abstract

Milk and dairy are rich in insulin-like growth factor 1 (IGF-1), a protein secreted through the action of growth hormone (GH) and implicated in growth and metabolism. Objective: This study aimed to investigate the roles of milk intake and body composition and identify the presence of the single nucleotide variant (SNV) rs6214 in the insulin-like growth factor 1 gene (*IGF1*) and its effects on the serum IGF-1 and GH levels and body composition. Methods: We analyzed 110 volunteers with and without a history of milk intake. Through a case–control study with one hundred ten healthy volunteers, serum IGF-1 and GH levels were measured using the ELISA technique, the body composition was determined with bio-electrical impedance equipment, genotyping of the rs6214 SNV was carried out using real-time PCR, and a dietary questionnaire was administered to assess milk intake, with or without consumption. Results: The results showed that the highest levels of IGF-1 were found in people who regularly consumed milk, along with a lower body mass index (BMI) and percentage of fat. A lower BMI and fat percentage were associated with higher levels of IGF-1, lean mass, and SNV presence. Lower levels of BMI and percentages of subcutaneous and visceral fat were found in regular milk consumers. Conclusions: Our study suggests that dairy intake and the IGF1 gene rs6214 SNV are associated with higher levels of IGF-1, high levels of lean mass, a low BMI, a low % fat, and low visceral fat.

## 1. Introduction

Mexico is among the top ten countries in obesity rates and the leading consumers of dairy products worldwide [1]. Dairy consumption has been associated with a normal body mass index (BMI = 18–24.9) [2], as well as lower levels of body fat and visceral fat, compared to individuals following a dairy-free diet [3]. Milk intake has been linked to elevated levels of plasma IGF-1 [4], as thermal treatment has been shown to have no influence on the concentrations of IGF-I and IGF-II in human milk and IGF-I levels in bovine colostrum are reduced by approximately 10% upon heating [5]. In low-fat milk and reduced-fat milk, IGF-1 levels are lower [4].

Insulin-like growth factor 1 (IGF-1), also known as somatomedin C, is a protein secreted through the action of growth hormone-binding protein (GHBP) [6,7,8]. It plays a role in growth and development, as well as metabolism. It has hypoglycemic and anabolic effects [8,9].

Ninety percent of circulating IGF-1 is of hepatic origin and exerts autocrine, paracrine, and endocrine effects on multiple tissues. Low serum levels of IGF-1 have been associated with an increased risk of chronic degenerative diseases such as type 2 diabetes, cancer, cardiovascular diseases, and neuropathy [10].

The function of IGF-1 stimulates skeletal muscle hypertrophy and a shift to glycolytic metabolism by activating calcium–calmodulin-dependent calcineurin phosphatase (calcineurin A; 114105) and inducing nuclear factor of activated T cells, cytoplasmic 1 protein (NFATC1) (600489), which is postulated as an important factor in obesity prevention [6,11], It is a significant risk factor for the development of multiple pathologies, such as metabolic and cardiovascular diseases, various neoplasms, neurodegenerative disorders, and musculoskeletal system alterations [12].

Additionally, IGF-1 is involved in skeletal muscle development by stimulating protein synthesis through the PI3K/AKT pathway, which is involved in protein synthesis and inhibits degradation [13]. It also participates in the regulation of growth and metabolism. It activates the Raf-1/MEK/ERK pathway, which is involved in cell proliferation, leading to increased energy expenditure and, consequently, a reduction in fat mass [14].

Although nutrition is considered a key factor in IGF-1 levels, the most significant variation of 40 to 60% depends on hereditary factors [15]. Single nucleotide variants (SNVs) in the IGF1 gene, such as the rs6214 C<T SNV, have been reported to be associated with elevated serum levels of IGF-1 [16,17]. The rs6214 is located in the three prime untranslated region (3′-UTR) wich contains and im-portant sequences that regulate mRNA transcription, stability, cellular localization, and microRNA binding [18].

## 2. Materials and Methods

### 2.1. Study Design

This is an observational prospective, case–control, association study approved by the research ethics committee of the “Hospital General Dr. Santiago Ramón y Cajal” ISSSTE (CONBIOETICA 10CE100120130723 and COFEPRIS 13 CEI 10005 128) and performed according to the Declaration of Helsinki and Mexican General Health Law. Each participant signed an informed consent letter.

### 2.2. Study Population

The sample size was calculated using the case–control hypothesis contrast formula, with a confidence interval of 95% and a reliability of 85%. The mutation frequency was considered based on the Latin American two-population report in dbSNP [19], the overweight frequency, and milk intake in the population from the state of Durango, which were taken as references.

Based on these considerations, 99 subjects were required to ensure our study’s representativeness and avoid the need for continuity correction in the X^2^ value. Therefore, a total sample size of 110 individuals was recruited, including 55 volunteers without milk intake and 55 volunteers with milk intake, aged between 20 and 59 years, both men (40.9%) and women (59.09%), apparently healthy, from the state of Durango. The participants were recruited from a public health institution, 450 General Hospital in the state of Durango, and belonged to the free public health program INSABI. The exclusion criteria for participating in the study were serious liver problems, use of steroids or thyroid hormones, pregnancy, and consumption of skim milk.

### 2.3. Methods

Milk consumption and exclusion criteria were obtained by questionnaires applied to volunteers. The questionnaire was validated using Cronbach’s alpha validation test [20]. The questionnaire included daily milk intake (Appendix A). The minimum amount of milk per day was 200 mL according to the provisions of the “Food and Physical Activity Guidelines for the Mexican Population” [21,22]. The minimum total consumption of cheese and yogurt was obtained from MyPyramid data, version 2.0 (MPED 2.0) [23].

### 2.4. Body Composition Evaluation

The variables analyzed were weight, BMI, and body composition (BC) elements such as the total fat percentage, visceral fat, and lean mass (bioimpedance analysis). Impedance was obtained through the body composition monitors OMRON HBF-516 and TANITA TBF300-A. The classification the BMI parameters established by the WHO was considered, where low weight corresponds to a BMI less than 18.5, normal weight = a BMI 18.5 to 24.9, overweight = a BMI 25.0 to 29.9, and obesity = a BMI greater than 30.0.

### 2.5. Non-Dietary Covariates

Data on other covariates in this study were obtained face-to-face through standardized questionnaires, which included demographic characteristics, lifestyle factors, and individual disease history (smoking, alcohol consumption, physical activity level, and whether the individual suffered from coronary heart disease, hypertension, or diabetes).

### 2.6. Blood Sample Collection

Blood samples were obtained by peripheral venipuncture from the patient’s forearm early in the morning after fasting for 8–10 h. A total of 4 mL was collected in vacutainer-type tubes containing EDTA and without anticoagulant, and kept refrigerated at 2 to 8 °C until transported to the laboratory.

### 2.7. Genotypification Assay

Genomic DNA was obtained from whole blood using the DTAB-CTAB method [24], and DNA integrity and purity were quantified and evaluated by nano spectrophotometry and electrophoresis. The IGF1 gene rs6214 SNV was analyzed by real-time PCR using the StepOne system™ (Applied Biosystems, Thermo Fisher Scientific Corporation, Foster City, CA, USA) and the TaqMan^®^ probe C__11495137_10 specific for the IGF1 gene rs6214 SNV (context sequence: TCACATCTAACTATGACAGAAAACA (C/T) GTTAAGTCTG-CAGAAGACTGCCTAT). The SNV was classified as mutated homozygous, wild-type homozygous, or heterozygous.

### 2.8. GH and IGF-1 Determination

The determination of GH and IFG-1 levels from blood serum was obtained using the Enzyme-Linked Immunosorbent Assay (ELISA) technique in “ELIREAD” model “ELISA Reader” equipment using a double antibody sandwich immunoassay and solid phase kits produced by the SIGMA-ALDRICH laboratory specifically for the determination of growth hormone (GH) (Human GH ELISA Kit RAB0206) and human IGF-1 (Human IGF-I ELISA Kit RAB0228) levels.

### 2.9. Statistical Analysis

Descriptive statistics were reported as means. The Chi-squared test was used for the statistical analysis to compare the variables. The odds ratio (OR) and the 95% confidence intervals were reported and were calculated using 2 × 2 tables by testing three genetic modes of inheritance, that is, dominant, codominant, and recessive models. High levels of IGF-1 were defined as those above the mean (169.4 ng/mL)—the cut-off value taken from Rosenthal and Rubin’s criteria [25,26]. The variables of age, BMI, visceral fat, lean mass, fat %, and IGF-1 were categorized to facilitate the statistical analysis. Differences were considered statistically significant if *p* < 0.05. The statistical software used was SPSS (IBM Corp. Released 2013. IBM SPSS Statistics for Windows, Version 23.0. Armonk, NY, USA: IBM Corp.) [27].

## 3. Results

### 3.1. Body Composition and GH and IGF-1 Levels

We characterized the distribution of the variables sex, age, BMI, weight, fat percentage, lean mass, viscera fat, GH, and IGF-1 levels concerning milk intake in each group (Table 1) as numbers or means and standard deviations. No significant differences concerning milk intake were found in age (*p* = 0.254, *p* = 0.345, *p* = 0.567, and *p* = 0.132). Neither were significant differences found for sex (*p* = 0.063 and *p* = 0.075) in milk intake. We did not find significant differences in the lean mass variables (*p* = 0.051, respectively). The variables that showed differences were BMI (*p* = 0.0003), fat percentage (*p* = 0.021), GH levels (*p* = 0.0001), IGF-1 levels (*p* = 0.0002), and visceral fat (*p* = 0.011). We found that the lowest BMI, fat percentage, and concentrations of GH were in the milk intake group, along with the highest levels of lean mass and IGF-1 concentrations.

Figure 1 shows a principal component analysis of the grouping of the variables that were associated with each other and that had similarity in their effects. It can be seen that the variables of lean mass, muscle, snv, igf.1, gh and milk intake were grouped, while the variables of weight, visceral fat, BMI, fat mass, fat and body fat generated another group.

### 3.2. Genotyping

Table 2 displays the genotypic and allelic frequencies of the SNV rs6214 in the IGF1 gene within the studied population. The frequency of the mutated allele (T) was determined to be 31%, while the wild-type allele accounted for 68%. This mutation has not been reported in the Mexican population.

The logistic univariate analysis indicated that low serum levels of IGF-1 were associated with an increased risk of chronic degenerative diseases between the IGF1 gene rs6214 C>T variant according to the modes of inheritance and the levels of GH and IGF-1, as shown in Table 2. It was observed that the presence of the mutated allele T was directly related to high IGF-1 levels in codominant, dominant, and recessive modes of inheritance [OR = 10.6, 95% CI (2.3–18.63), *p* = 0.0021], [OR = 3.41, 95% CI (1.31–8.56), *p* = 0.008], [OR = 6.75, 95% CI (1.57–28.9), *p* = 0.009], respectively. Additionally, milk intake was directly associated with high levels of IGF-1 [OR = 17.09, 95% CI (3.63–19.10), *p* = 0.0003]. There was no association between the presence of the IGF1 gene rs6214 C>T SNV and the levels of growth hormone (GH).

Table 3 presents the correlation analysis between milk intake, BMI, GH levels, and IGF-1 levels with the fat percent, lean mass, BMI, visceral fat, and milk intake. The results revealed a negative correlation between milk intake, BMI, fat percentage, and visceral fat. Conversely, a positive correlation was observed between milk intake, IGF-1 levels, and lean mass. These findings indicate that milk consumption increases IGF-1 levels above the average. Negative correlations were observed between IGF-1 levels and both fat percentage and BMI.

GH levels correlated negatively with milk intake and the fat percentage, and a positive correlation was observed between BMI, lean mass, and visceral fat.

The logistic analysis of the association between an individual’s obesity and body composition with GH and IGF-1 levels is shown in Table 4. It was observed that GH levels were not clearly associated with visceral fat [OR = 0.6, 95% CI (0.11–3.02), *p* = 0.022], and IGF-1 levels showed a direct association with high percentages of lean mass [OR = 1.10, 95% CI (1.45–2.68), *p* = 0.014].

The associations between the rs6214 SNV and body composition parameters are shown in Table 5. Statistically significant associations (*p* < 0.05) were observed between the presence of the mutated allele and lean mass (*p* = 0.016) and BMI (*p* = 0.029). No significant associations were found between the percentage of fat (*p* = 0.41) and visceral fat (*p* = 0.28).

In codominant and recessive models, the presence of the TT genotype (mutated homozygote) of rs6214 was associated with BMI values (18–24.9) within normal weight parameters [OR = 0.19; 95% CI (0.05–0.68), *p* = 0.002], as shown in Table 6.

## 4. Discussion

The present study highlights that body composition parameters such as BMI, fat mass, and lean mass differ significantly between the two study groups. These differences may be partially attributed to demographic factors and dietary habits, which are known to influence metabolic profiles, like milk intake, which has been previously associated with improved body composition outcomes.

A possible reason for the clustering of lean mass, muscle, IGF-1 levels, GH levels, and the SNV lies in the anabolic nature of IGF-1; this hormone not only helps prevent a decrease in muscle mass but also promotes greater energy consumption and therefore an increase in and the development of muscle mass. It is important to highlight that the synthesis and secretion of IGF-1 are directly influenced by the presence of growth hormone (GH) in the blood circulation [28]. In addition to this, it has been reported that the presence of the SNV rs6214 increases IGF-1 levels, and so its grouping would be related to this behavior [29].

Our findings align with prior studies suggesting that certain components in milk, such as proteins [30], calcium [31], and conjugated linoleic acid (CLA) [32], may promote a favorable body composition by stimulating lipolysis and inhibiting lipogenesis [33], raising HDL cholesterol levels, reducing LDL cholesterol levels, and enhancing energy expenditure, ultimately reducing fat mass through decreased adipocyte proliferation [34].

A key observation from this study is the positive correlation between milk intake and elevated IGF-1 levels (r = 0.237, *p* = 0.002). This is supported by the calculated odds ratio [OR = 17.09, 95% CI (3.63–19.10), *p* = 0.003], indicating a strong association between milk intake and IGF-1 levels above the population mean. The previous literature reports that milk consumption in infants (>500 mL/day) increases IGF-1 concentrations by 9–20% [13] and in adults by approximately 10% [16].

One hypothesis for this effect is the structural similarity between bovine and human IGF-1, which may permit its intact absorption, as evidenced in rodent models [35]. A study carried out by Romo Ventura and collaborators in a population of healthy adult men and women aged 18 to 80 years demonstrated that milk consumption increased IGF1 levels. They demonstrated that an increase in milk consumption of 200 and 400 g per day increased IGF-1 levels compared to people who did not consume milk [36].

Interestingly, we also identified a negative correlation between BMI and IGF-1 levels above the mean (r = −0.147, *p* = 0.056). This finding is consistent with studies that have associated low IGF-1 levels with an elevated BMI, suggesting an inverse relationship between circulating IGF-1 levels and adiposity [37].

Moving beyond dietary factors, our study explored the influence of genetic variation, particularly the rs6214 single nucleotide variant (SNV) in the *IGF1* gene, on IGF-1 levels and body composition. The rs6214 polymorphism showed linear correlations with elevated IGF-1 levels and increased lean mass. Individuals carrying at least one mutated allele demonstrated higher serum IGF-1 concentrations, regardless of the inheritance model. 

This SNV is located in the 3′ untranslated region (3′-UTR) of IGF1, which, although it does not alter the amino acid sequence, may exert post-transcriptional regulatory functions that increase IGF1 mRNA stability and translation [6,16,17].

The interaction between rs6214 and milk intake appears to have a synergistic effect. Our data suggest that this interaction is associated with healthy BMI values (<24.9) and increased lean mass [OR = 2.45, 95% CI (1.08–5.53)].

IGF-1 plays a critical role in skeletal muscle protein synthesis and growth regulation [38], especially when both a genetic predisposition and nutritional stimuli are present [39,40,41]. Velloso [42] also documented this interaction, highlighting the combined effects of IGF-1 levels, milk intake, and muscle mass gain. Regarding the allelic distribution, the frequency of the mutated rs6214 allele (T) in our study population was 31%, and 68% for the wild-type allele.

This is a novel report for the Mexican population. The most comparable data come from the Mexican-American population, with frequencies of 40.6% (mutated) and 59.3% (wild-type) [43]. Other populations, such as those in Peru, show a 27% frequency for the mutated allele [43]. European populations, including those in Italy, the UK, and Spain, report similar frequencies (40.1%, 39%, and 41.1%, respectively), whereas the East Asian population demonstrates a higher frequency of the mutated allele (52%) [44,45,46].

It is well established that SNVs can influence nutrient–gene interactions, modulating metabolic outcomes such as BMI, fat mass, and lean mass. In this context, dairy intake, particularly milk and yogurt, has been associated with lower prevalences of overweight and obesity, a decreased BMI, reduced waist circumference, and thinner subscapular skinfolds [47,48,49]. In our cohort, whole milk intake was significantly associated with higher levels of both IGF-1 and GH in Mexican individuals.

Despite these relevant findings, our study has limitations. First, we only measured serum IGF-1 levels after dairy intake, without a baseline measurement prior to consumption. Future studies should include pre- and post-consumption serum assessments to better delineate the direct effect of milk on IGF-1 levels [50,51,52,53,54,55].

Second, we did not establish a minimum effective dose of dairy products required to induce a measurable increase in IGF-1 levels, which limits the clinical application of our findings.

Addressing these limitations could strengthen causal interpretations. Finally, a plausible biological mechanism for our observations is the activation of the Raf/MEK/ERK signaling cascade by IGF-1, which leads to enhanced cell proliferation and energy expenditure, contributing to a reduction in the adipose tissue mass [56].

## 5. Conclusions

In this study, the results obtained show that individuals who have the mutated allele (T) in the C>T SNV of the IGF1 rs6214 gene and regular consumption of milk have higher levels of IGF-1 but not of GH, which is related to high levels of lean mass and low levels of BMI, % fat, and visceral fat. These results suggest that the integration of milk and dairy products represents potential support for the dietary management of obesity. These results indicate that the presence of the rs6214 variant of the IGF1 gene in combination with milk intake promises better results in the treatment of obesity. Therefore, it is important to consider conducting clinical trials with diets that include milk intake in obese patients.

## Figures and Tables

**Figure 1 metabolites-15-00556-f001:**
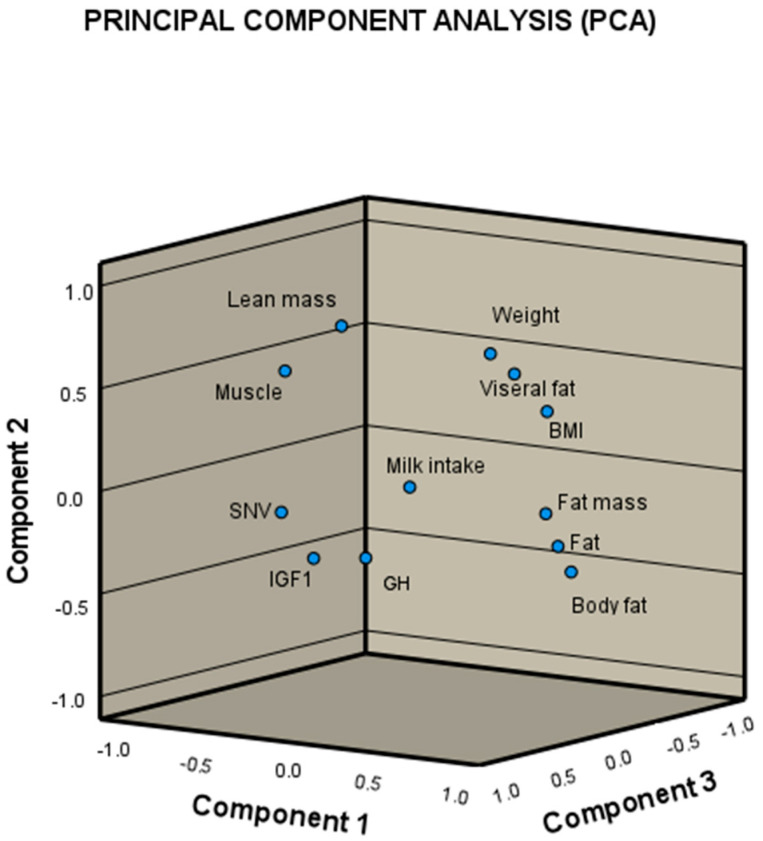
Principal component analysis.

**Table 1 metabolites-15-00556-t001:** Comparative analysis of the differences between groups based on milk intake.

Variable	Milk Intake	Without Milk Intake	*p*-Value X^2^
BMI (Kg/m^2^) > 25, N	18	37	0.0003
Fat percentage > 30, N	22	34	0.021
Lean mass > 35, N	19	10	0.051
Visceral fat > 5, N	32	44	0.011
GH (ng/mL), Mean ± SD	0.698 ± 0.08	0.793 ± 0.16	0.0001
IGF-1 (ng/mL) Mean ± SD	224.5 ± 50.5	118.4 ± 32.8	0.0002
Age (years)			
20–29	22	20	0.254
30–39	12	12	0.345
40–49	11	12	0.567
50–59	10	11	0.132
Sex			
Male	25	20	0.063
Female	30	35	0.075

SD = standard deviation; BMI = body mass index; GH = growth hormone; IGF-1 = insulin-like growth factor 1.

**Table 2 metabolites-15-00556-t002:** Logistic analysis of the associations between GH and IGF-1 levels in relation to milk intake and the presence of *IGF1* gene C>T rs6214 variant.

	GH			IGF-1		
	OR	CI_95_	*p*-Value	OR	CI_95_	*p*-Value
Milk intake No		Ref.			Ref.	
Yes	1.04	0.49–2.18	0.904	17.09	3.63–19.10	0.0003
Codominant model CC		Ref.			Ref.	
CT	2.85	0.7–11.40	0.13	2.5	0.9–6.68	0.065
TT	2.65	0.33–14.45	0.33	10.6	2.3–18.63	0.0021
Dominant model CC		Ref.			Ref.	
TT + CT	1.85	0.8–6.42	0.10	3.41	1.31–8.56	0.008
Recessive model CC + CT		Ref.			Ref.	
TT	1.89	0.22–16.05	0.55	6.75	1.57–28.9	0.009
C Allele		Ref.			Ref.	
T Allele	2.20	0.7–6.18	0.12	3.05	1.5–5.9	0.001

GH = growth hormone; IGF-1 = insulin-like growth factor 1; OR = odds ratio; CI = confidence interval.

**Table 3 metabolites-15-00556-t003:** Correlation analysis between milk intake, BMI, GH levels, and IGF-1 levels with fat %, lean mass, BMI, visceral fat, and milk intake.

Variables	Milk Intake		BMI		Fat %		Lean Mass		Visceral Fat	
		*p*-Value	r	*p*-Value	r	*p*-Value	r	*p*-Value	r	*p*-Value
Milk intake		1	−0.278	0.000	−0.210	0.006	0.101	0.190	−0.117	0.130
BMI	−0.278	0.000	1		0.480	0.000	0.195	0.011	0.648	0.000
GH	−0.042	0.588	0.101	0.193	−0.018	0.817	0.101	0.193	0.053	0.494
IGF-1	0.237	0.002	−0.147	0.056	−0.102	0.187	0.065	0.040	0.038	0.620

GH = growth hormone; IGF-1 = insulin-like growth factor 1; BMI = body mass index; r = Pearson’s correlation coefficient.

**Table 4 metabolites-15-00556-t004:** Logistic analysis of the associations between an individual’s obesity and body composition with GH and IGF-1 levels.

	Total Fat Mass			Lean Mass			Visceral Fat			Obesity		
	OR	CI_95_	*p*-Value	OR	CI_95_	*p*-Value	OR	CI_95_	*p*-Value	OR	CI_95_	*p*-Value
GH	2.18	0.60–7.82	0.22	0.68	0.19–2.34	0.54	0.6	0.11–3.02	0.022	0.66	0.15–2.80	0.57
IGF-1	0.68	0.28–1.65	0.39	1.10	1.45–2.68	0.014	0.69	0.26–1.83	0.45	0.62	0.25–1.52	0.19

GH = growth hormone; IGF-1 = insulin-like growth factor 1; OR = odds ratio; CI = confidence interval.

**Table 5 metabolites-15-00556-t005:** Body composition of the participants in relation to the presence of the rs6214 SNV.

	BMI > 25	% Fat	Lean Mas	Visceral Fat
	*X* ^2^	*X* ^2^	*X* ^2^	*X* ^2^
	*p* Value	*p* Value	*p* Value	*p* Value
CC	Reference	Reference	Reference	Reference
CT	0.154	0.90	0.025	0.48
TT	0.0087	0.033	0.12	0.17
TT + CT	0.029	0.41	0.016	0.28

BMI = body mass index; CC = wild homozygote; TT = homozygous mutated; CT = heterozygous.

**Table 6 metabolites-15-00556-t006:** Analysis of the genotype risk according to the inheritance model with respect to BMI.

Model	Genotype	<Median BMI	>Median BMI	OR	IC_95%_	*p* Value
CO	IGF-1					
	rs6214					
	CC	27 (48)	28 (51.02)	1		------------
	CT	18 (32)	23 (44.89)	1.32	0.55–3.12	>0.05
	TT	10(20)	3 (4.08)	0.19	0.05–0.68	0.002
	Allele C	69 (62.7)	78 (70.9)	1		-----------
	Allele T	41 (37.3)	32 (29.1)	0.64	0.34–1.18	>0.05
DO	CC	25 (48)	28 (51.02)	1		
	CT-TT	30 (52)	27 (48.97)	0.88	0.37–2.08	>0.05
RE	CC-CT	43 (80)	50 (95.91)	1		
	TT	12 (20)	5 (4.08)	0.17	0.04–0.71	0.003

OR = odds ratio; IC = confidence interval; CO = codominant; DO = dominant; RE = recessive; BMI = body mass index.

## Data Availability

The raw data supporting the conclusions of this article will be made available by the authors upon request due to privacy.

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
