# Peer review of "Effects of IGF1 rs6214 Polymorphism and Milk Consumption on Serum Levels of IGF-1 and GH and Body Composition"

_metabolites, 2025, doi:10.3390/metabo15080556_

Round 1

Reviewer 1 Report

Comments and Suggestions for Authors

This study explores the relationship between milk intake, the IGF1, rs6214, SNV, and their combined influence on IGF-1/GH levels and body composition. The topic is novel and relevant, addressing the interaction between nutrition and genetics. The manuscript is generally well organized, the statistical approach is sound, and the data is adequately presented. However, several minor revisions are suggested to improve clarity and overall presentation.

  1. The minimum intake of 200 mL/day is mentioned. However, the discussion lacks an explanation of whether there is a dose-dependent relationship with IGF-1 levels. Consider adding a comment on whether the physiological effect increases linearly or if a threshold effect is suspected.
  2. The genotyping of IGF1 rs6214 is appropriately conducted, but the biological plausibility of its role could be strengthened. It is recommended to cite literature supporting the known or hypothesized regulatory function of rs6214, such as whether it is located in a promoter region or 3’UTR, and whether it has been previously linked to IGF-1 expression or metabolic traits.
  3. The study sample comprises healthy adults, but the manuscript lacks details on age distribution, sex balance, or covariate adjustment. It is unclear whether participants with lactose intolerance or specific dietary patterns were excluded. Additionally, variables such as physical activity, baseline diet, and socioeconomic status were not described. Including this information would enhance the accuracy of interpretation.

Author Response

  1. The minimum intake of 200 ml/day is mentioned. However, the discussion lacks an explanation of whether there is a dose-dependent relationship with IGF-1 levels. Please consider adding a comment on whether the physiological effect increases linearly or if a threshold effect is suspected.

Response: We have added documentary information showing that intake was measured and reflected in IGF-1 levels (lines 360-364).

  1. IGF-1 rs6214 genotyping was performed correctly, but the biological plausibility of its function could be strengthened. It is recommended to cite literature that supports the known or hypothesized regulatory function of rs6214, for example, whether it is located in a promoter region or 3'UTR, and whether it has been previously linked to IGF-1 expression or metabolic characteristics.

Response: The function and biological plausibility were strengthened by integrating information from a new reference (lines 89-91).

  1. The study sample is composed of healthy adults, but the manuscript lacks details regarding age distribution, gender balance, or covariate adjustment. It is unclear whether participants with lactose intolerance or specific dietary patterns were excluded. Furthermore, variables such as physical activity, baseline diet, or socioeconomic status were not described. Including this information would improve the accuracy of interpretation.

Response: The requested information on lines 114-116 is redacted.

Reviewer 2 Report

Comments and Suggestions for Authors

Some issues should be addressed before acceptance.

  1. All abbreviations should be full name when they firstly appeared.
  2. The title should be revised to reflect the contents of manuscript more clearly.
  3. Please check if the name of genes should be italic.
  4. The structure of introduction should be arranged more legitimately. There really is no need to divide it into so many paragraphs.
  5. Authors should specify that GH、IGF-1、SNV rs6214 were from serum or blood, and the style of terms should be consistent throughout the entire text.

Author Response

  1. All abbreviations should be the full name as they first appeared.

Answer: The full name is included before all abbreviations in their first citation (Lines 25, 26, 29, 30, 31, and 40).

  1. The title should be revised to more clearly reflect the content of the manuscript.

Answer: The title has been modified, and we have added more detailed wording than what was previously done.

  1. Please check whether gene names should be italicized.

Answer: The entire document has been reviewed and verified that only genes are mentioned in italics, not only the protein itself.

  1. The structure of the introduction should be more clearly organized. There is really no need to divide it into so many paragraphs.

Answer: We tried to avoid long prose for a better understanding of the document.

  1. Authors should specify that GH, IGF-1, and SNV rs6214 are derived from serum or blood, and the style of the terms should be consistent throughout the text.

Answer: The source of each GH, IGF-1, or SNV component is specified (lines 158-159 and 161).

Reviewer 3 Report

Comments and Suggestions for Authors

This article investigates the combined effects of milk intake and the insulin-like growth factor 1 (IGF-1) gene rs6214 single nucleotide variant on body composition and serum levels of IGF-1 and growth hormone (GH) in the Mexican population, providing new insights into how nutrition and genes jointly influence human metabolism and body composition. Some detailed issues need to be addressed before resubmitting it for review. My section-specific comments are listed below.

Abstract

- When a professional term first appears, please write out its full name and indicate the abbreviation in parentheses immediately following it.

- Line25: “Dairy” should be changed to “Dairy products”, “by” should be changed to “through”

- Line26, 27: “The study aimed to found role of milk intake” should be changed to “The study aimed to investigate the role of milk intake”

- Line44: “rs6412” should be changed to “rs6214”, Please maintain consistency throughout the full text.

- It is recommended to revise and polish the structure and content of the abstract.

Introduction

- Line49: “maximum range body mass index (BMI= 18-24.9)” should be changed to “normal-range body mass index (BMI= 18-24.9)”

Materials and methods

- Line133: “in the patient’s forearm” should be changed to “from the patient’s forearm”

- Please elaborate and supplement all relevant methodological details and quality control measures.

Results

- Line176: “show” should be changed to “showed”

- Table1-6: It is recommended to unify the number of decimal places.

Discussion

- The depth and breadth of the discussion are insufficient.

Author Response

  1. Abstract - When a professional term appears for the first time, write its full name and indicate the abbreviation in parentheses immediately afterward.

Answer: The full name is included before all abbreviations in their first citation (lines 25, 26, 29, 30, 31 and 40).

  1. Line 25: “Dairy” should be changed to “Dairy products.”

Answer: Since not all dairy products were included, but only whole milk, the phrase “milk intake” is included instead of dairy products (Line 28).

  1. “by” should be changed to “through.”

Answer: The suggested change is made (Line 32).

  1. Lines 26 and 27: “The study aimed to determine the role of milk intake” should be changed to “The study aimed to investigate the role of milk intake.”

Answer: The objective is reworded (lines 27-31).

  1. Line 44: “rs6412” should be changed to “rs6214.” Maintain consistency throughout the text.

Answer: The correction is made (Line 47).

  1. It is recommended that the structure and content of the abstract be reviewed and refined.

Answer: It is restructured and rewritten more explicitly, following the required structure. (Lines 25-46)

  1. Line 49: “Maximum body mass index (BMI = 18-24.9)” should be changed to “Normal body mass index (BMI = 18-24.9).”

Response: The wording is corrected (Line 52).

  1. Materials and methods. Line 133: “on the patient's forearm” should be changed to “from the patient's forearm.”

Response: The suggested change is made (Line 144).

  1. Please develop and supplement all relevant methodological details and quality control measures.

Answer: The BMI classification is detailed (Lines 132-136), information about the sample collection, quantity, temperatures before transfer to the laboratory is added (Lines 145 to 148), rs6214 classification (Lines 158-159), and the ELISA kits used are added (Lines 166-167).

  1. Results Line 176: “show” must be changed to “showed”.

Answer: The requested change is made (Line 190)

  1. Table 1-6: It is recommended to unify the number of decimal places.

Response: Unify to 3 digits.

  1. Discussion: The depth and breadth of the debate are insufficient.

Response: The discussion is expanded (Lines 360-364).

Round 2

Reviewer 3 Report

Comments and Suggestions for Authors

The authors have responded carefully to my comments, and have revised the manuscript accordingly, now the article is ready to be accepted.